# Conserved allosteric pathways for activation of TRPV3 revealed through engineering vanilloid-sensitivity

**Feng Zhang[1,2]\*, Kenton Jon Swartz[1], Andres Jara-Oseguera[1]\***

[1]Molecular Physiology and Biophysics Section, Porter Neuroscience Research Center, National Institute of Neurological Disorders and Stroke, National Institutes of Health, Bethesda, United States; [2]Department of Biochemistry, University of Utah, Salt Lake City, United States

**Abstract** The Transient Receptor Potential Vanilloid 1 (TRPV) channel is activated by an array of stimuli, including heat and vanilloid compounds. The TRPV1 homologues TRPV2 and TRPV3 are also activated by heat, but sensitivity to vanilloids and many other agonists is not conserved among TRPV subfamily members. It was recently discovered that four mutations in TRPV2 are sufficient to render the channel sensitive to the TRPV1-specific vanilloid agonist resiniferatoxin (RTx). Here, we show that mutation of six residues in TRPV3 corresponding to the vanilloid site in TRPV1 is sufficient to engineer RTx binding. However, robust activation of TRPV3 by RTx requires facilitation of channel opening by introducing mutations in the pore, temperatures > 30°C, or sensitization with another agonist. Our results demonstrate that the energetics of channel activation can determine the apparent sensitivity to a stimulus and suggest that allosteric pathways for activation are conserved in the TRPV family.

DOI: https://doi.org/10.7554/eLife.42756.001

\*For correspondence:
feng.zhang2@nih.gov (FZ);
andres.jara-oseguera@nih.gov (AJ)

## Introduction

Transient receptor potential (TRP) cation channels are involved in a diverse array of physiological functions (*Li, 2017*), with many of them acting as sensory detectors of stimuli such as temperature, natural products and various cell-signaling molecules (*Flockerzi, 2007*). Structurally, TRP channels are tetramers, with each subunit containing six transmembrane helices (S1-S6); the region encompassing the S5 through S6 helices forms the central ion-conducting pore, and the S1-S4 helices of each subunit form voltage sensor-like domains located peripherally in a domain-swapped arrangement (*Madej and Ziegler, 2018*).

Consistent with the remarkable diversity of biological roles that TRP channels play, the types of stimuli that activate these channels also largely differ between TRP subtypes, even among closely related members of the same TRP channel subfamily (*Flockerzi, 2007*; *Li, 2017*). However, many important structural features are highly conserved within each TRP channel subfamily, and between TRP channels in general (*Palovcak et al., 2015*; *Bae et al., 2018*; *Kasimova et al., 2018*; *Madej and Ziegler, 2018*; *McGoldrick et al., 2018*; *Zheng et al., 2018a*; *Zheng et al., 2018b*; *Zubcevic et al., 2018a*; *Zubcevic et al., 2018b*), raising the possibility that TRP channels that are sensitive to different stimuli share mechanisms of activation downstream of their ligand-interaction sites. This is well exemplified by TRPV1 and its closest homologue, TRPV2 (48.4% of sequence identity). TRPV1, an integrator of pain-producing stimuli in nociceptors (*Moore et al., 2018*), is the only member of the TRPV (vanilloid) subfamily of TRP channels that can be activated by vanilloid compounds (*Caterina et al., 1997*; *Jordt and Julius, 2002*), cysteine-reactive molecules (*Salazar et al., 2008*), and some agonists that directly interact with the extracellular face of the pore domain, such

as protons (*Tominaga et al., 1998*; *Jordt et al., 2000*) and the double-knot toxin (DkTx) from tarantula venom (*Bohlen et al., 2010*; *Cao et al., 2013*; *Bae et al., 2016*; *Gao et al., 2016*). In contrast, TRPV2 is activated by very few known stimuli, including heat (*Caterina et al., 1999*; *Yao et al., 2011*), the non-selective channel modulator 2-aminoethoxydiphenyl borate (2-APB) (*Hu et al., 2004*) and cannabinoid compounds (*Qin et al., 2008*), all of which also influence the activity of TRPV1 (*Hu et al., 2004*; *Qin et al., 2008*). In this context, it is remarkable that only four mutations in TRPV2 (TRPV2-4M) are sufficient to enable robust activation by the TRPV1-specific vanilloid agonist resiniferatoxin (RTx) (*Yang et al., 2016*; *Zhang et al., 2016*; *Zubcevic et al., 2018b*), suggesting that the sites where vanilloids interact with TRPV1 and TRPV2-4M are very similar, and that these channels share mechanisms of activation. Indeed, structural studies have confirmed that RTx binds to the same site between the S1-S4 domains and the pore domain in both TRPV1 (*Cao et al., 2013*; *Gao et al., 2016*) and TRPV2-4M (*Zubcevic et al., 2018b*) (*Figure 1—figure supplement 1A*).

In the present study, we explored whether vanilloid sensitivity could be engineered into the TRPV3 channel, which shares 42% sequence identity with TRPV1. TRPV3 is expressed primarily in mammalian keratinocytes, where it is required for the formation of the skin barrier (*Cheng et al., 2010*), and mutations in the channel can lead to skin diseases such as Olmsted Syndrome (*Lin et al., 2012*; *Zhi et al., 2016*). TRPV3 is a polymodal detector of warm temperatures (*Peier et al., 2002*; *Smith et al., 2002*; *Xu et al., 2002*), chemicals such as 2-APB (*Chung et al., 2004*; *Hu et al., 2004*) and multiple natural products such as camphor, eugenol and thymol (*Xu et al., 2006*; *Vogt-Eisele et al., 2007*). To probe for RTx sensitivity in TRPV3, we mutated six residues in mouse TRPV3 (TRPV3-6M) corresponding to the vanilloid-binding pocket of rat TRPV1, and found that although RTx can bind to the mutant channel, activation of TRPV3-6M by RTx requires facilitation of the opening transition, either by introducing mutations in the pore domain, or sensitizing the channel with heat or 2-APB. Our results provide functional evidence for conserved mechanisms of activation within the TRPV family, while also providing important information on TRPV3-specific activation properties.

## Results

### Engineering vanilloid sensitivity into the TRPV3 channel

We began by testing whether RTx sensitivity could be engineered into the TRPV3 channel. The vanilloid-binding pocket region is highly similar in TRPV1, TRPV2 and TRPV3, both in terms of amino acid sequence conservation (*Figure 1A*) and 3-dimensional structure (*Cao et al., 2013*; *Gao et al., 2016*; *Huynh et al., 2016*; *Zubcevic et al., 2016*; *Singh et al., 2018*; *Zubcevic et al., 2018b*) (*Figure 1—figure supplement 1A*), and four mutations are sufficient to confer RTx sensitivity to the rat, mouse and rabbit TRPV2 channels (*Yang et al., 2016*; *Zhang et al., 2016*; *Zubcevic et al., 2018b*). We initially introduced four mutations into mouse TRPV3 that are equivalent to those in rat TRPV2-4M to generate the TRPV3-4M channel (F522S/L557M/A560T/Q580E; *Figure 1A* and *Figure 1—figure supplement 1A*), expressed the construct in *Xenopus laevis* oocytes and used the two-electrode voltage-clamp technique to test for responses to RTx and to the non-specific TRPV channel agonist 2-APB (*Hu et al., 2004*). We recorded current time courses for channel activation at room temperature by each agonist (see *Figure 1B*, left panel and Materials and methods), and obtained current-voltage (I-V) relations in the absence and presence of activators, or in the presence of ruthenium red (RR), a non-specific inhibitor of TRP channels (see *Figure 1B*, right panel and Materials and methods). Whereas TRPV2-4M displayed robust responses to RTx (*Figure 1C*), as we showed previously (*Zhang et al., 2016*), TRPV3-4M was activated by 2-APB but not by 100 nM RTx (*Figure 1D*), a concentration that is saturating for both TRPV1 and TRPV2-4M (*Zhang et al., 2016*). We found two additional positions within the vanilloid pocket where the size, shape and polarity of the residues in TRPV3 largely differ from those in TRPV1 (H523 and W521 in TRPV3 correspond to E513 and Y511 in TRPV1, respectively; *Figure 1A* and *Figure 1—figure supplement 1A*). We therefore mutated these two additional positions in TRPV3-4M to the corresponding residues in TRPV1 to generate TRPV3-6M, and found that the construct remained insensitive to RTx, although it retained robust responses to 2-APB (*Figure 1E*).

The binding of an agonist to its site requires functional coupling to the pore domain to promote opening of the ion conduction pathway. To test whether differences in the pore domain might play

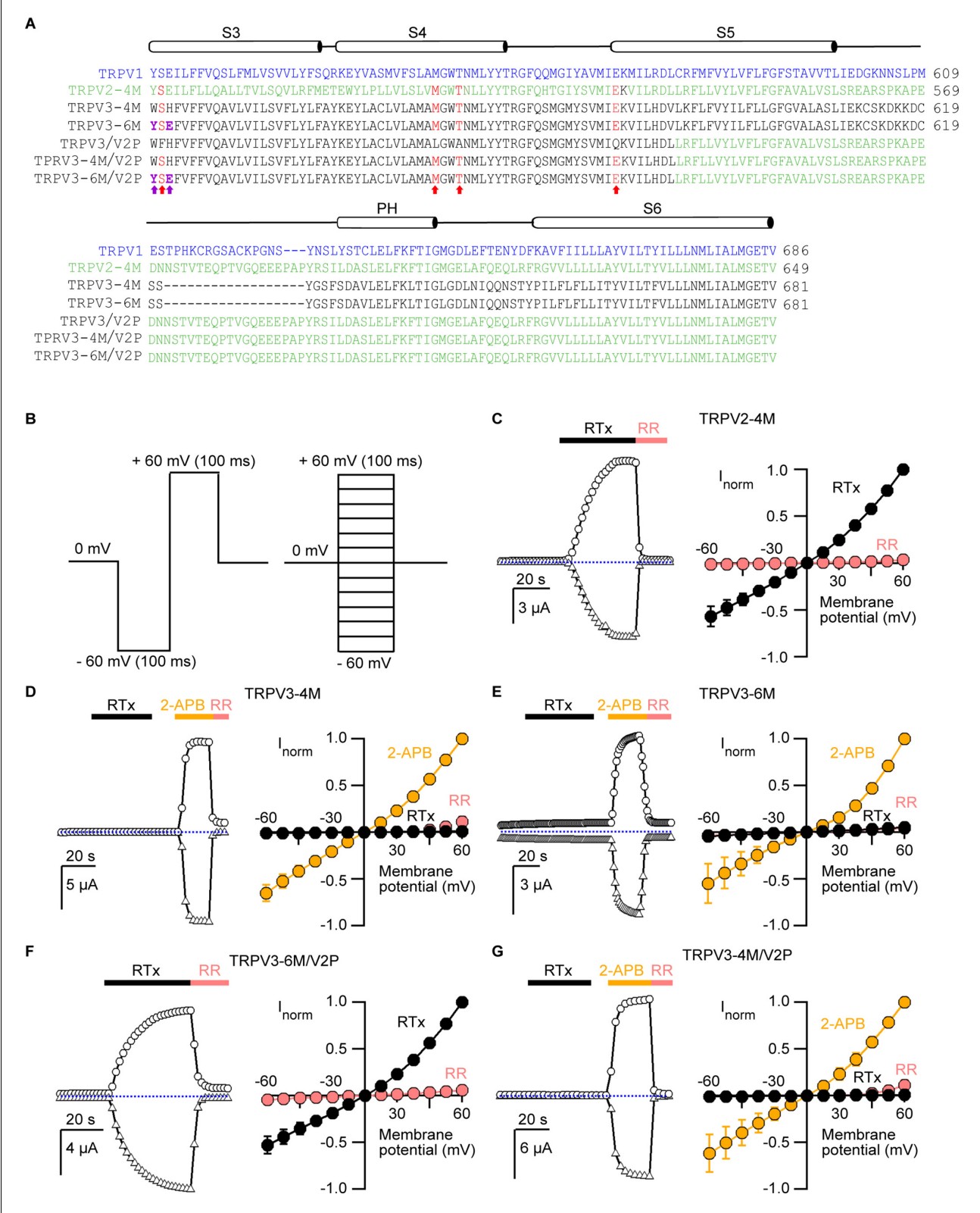

**Figure 1.** Engineering an RTx-binding site into the TRPV3 channel. (**A**) Sequence alignment of S3 through S6 TM helices of rat TRPV1 (blue), rat TRPV2-4M (F472S/L507 M/S510T/Q530E, green), mouse TRPV3-4M (F522S/L557M/S560T/Q580E, black), mouse TRPV3-6M (W521Y/H523E/F522S/L557M/A560T/Q580E, black) and mouse TRPV3 chimeras containing the pore domain of rat TRPV2 (TRPV3/V2P). Key residues involved in RTx binding in TRPV2-4M and TRPV3-4M that were mutated to the corresponding residues in TRPV1 are shown in red, with the two additional mutations (W521Y/

*Figure 1 continued on next page*

*Figure 1 continued*

H523E) in TRPV3-6M shown in purple and bold. (B) Voltage protocols used to measure time courses for activation (left) or I-V relations (right). In both cases voltage steps were elicited every 2 s. (C-G, left panel) Representative time courses of activation in response to RTx (100 nM) or 2-APB (3 mM) measured at ±60 mV. 50 µM ruthenium red (RR) was applied at the end of each recording to inhibit the channel. The dotted horizontal line indicates the zero-current level. The thick colored horizontal lines indicate the application of agonists or RR. (C-G, right panel) Mean normalized I–V relations obtained in the presence of 100 nM RTx (black), 3 mM 2-APB (yellow) and 50 µM RR (red-filled symbols). Currents were normalized to the current value in the presence of a saturating concentration of 2-APB (3 mM) or RTx (100 nM) at +60 mV. Data are expressed as mean ± S.E.M (n = 4–8 cells).

DOI: https://doi.org/10.7554/eLife.42756.002

The following figure supplement is available for figure 1:

**Figure supplement 1.** The vanilloid-binding pocket in the TRPV family.
DOI: https://doi.org/10.7554/eLife.42756.003

a role in preventing RTx activation in TRPV3-6M, we transferred the pore domain of rat TRPV2 into the TRPV3-6M construct to generate TRPV3-6M/V2P (*Figure 1A*). Remarkably, the resulting chimera was robustly activated by RTx and readily blocked by RR (*Figure 1F*), indicating that the pore domain of TRPV2 contains determinants that are important for RTx-activation. We also found that the two additional residues in the vanilloid pocket differing between TRPV3-4M and TRPV3-6M were necessary for RTx activation, since we transferred the pore domain of TRPV2 into TRPV3-4M to generate TRPV3-4M/V2P (*Figure 1A*), and found that the resulting chimera could be activated by 2-APB but not by RTx (*Figure 1G*).

## Mutations within the pore domain required for vanilloid activation of TRPV3

We next sought to identify the specific residues within the pore domain of TRPV2 that enable activation of TRPV3-6M/V2P by RTx. We aligned the sequences of TRPV1, TRPV2 and TRPV3 and found 21 amino acids in the pore domain that are conserved between TRPV1 and TRPV2 but differ in TRPV3 (*Figure 2A*). We individually mutated each of the 21 residues in the TRPV3-6M background to those in TRPV1 and TRPV2, and tested for their response to RTx and 2-APB. We identified five positions that when individually mutated resulted in constructs with robust RTx-activated currents that were very similar in magnitude to those elicited by 2-APB (V587L, A606V, F625L, F656I, F666Y; *Figure 2B,C*, dark blue). The TRPV3-6M + V587L construct exhibited a higher apparent RTx to 2-APB current amplitude ratio resulting from 2-APB-dependent desensitization after RTx-stimulation. Notably, these five mutated residues are widely distributed within the pore domain, located in the S5 helix (V587L and A606V), pore turret (F625L), and S6 helix (F656I and F666Y) (*Figure 2D* and *Figure 2—figure supplement 1*). Most of these residues (V587, F625 and F656) are located at the protein/lipid interface, close to the RTx-binding pocket in TRPV1 and TRPV2-4M (*Figure 2—figure supplement 1A–C*) (*Gao et al., 2016*; *Zubcevic et al., 2018b*). A606 is tightly packed between the S5 helix and the pore helix of the same subunit (*Figure 2—figure supplement 1A,B*). F666 is facing the ion-conduction pathway, and is located at a key position in the S6 helix proposed to function as a gating hinge in TRPV channels (*Palovcak et al., 2015*; *Zubcevic et al., 2016*; *Kasimova et al., 2018*; *McGoldrick et al., 2018*) (*Figure 2—figure supplement 1B,D*). In addition, five constructs with individual mutations at additional positions exhibited detectable responses to RTx (I595L, K611D, L635F, L639M, Y650F; *Figure 2B–D*, light blue, and *Figure 2—figure supplement 1*) that were, however, smaller than the responses to 2-APB in those same constructs. The remaining 11 individual residue substitutions we tested, which largely localize to positions in extracellular pore loops that are less conserved between TRPV channels (*Figure 2D* and *Figure 2—figure supplement 1A*), resulted in RTx-insensitive constructs that were still activated by 2-APB (L596V, L599F, S613K, D618P, S620E, I644F, Q646E, P651K, L653V, I659L, T660A; *Figure 2B,C*, grey).

Interestingly, some residue substitutions that conferred RTx sensitivity to TRPV3-6M did not involve large side-chain modifications, such as V587L and F666Y, whereas some mutations expected to be more disruptive based on more radical changes in side chain size or polarity, such as D618P, S620E or P651K, did not have an apparent effect on RTx- or 2-APB-senstivity (*Figure 2B,C*), possibly due to their location in more flexible and less conserved regions of the pore. Out of all substituted residues that conferred RTx responsiveness to TRPV3-6M, only the side-chain of V587 is close enough to the bound RTx to directly interact with the agonist, judging from the distances between

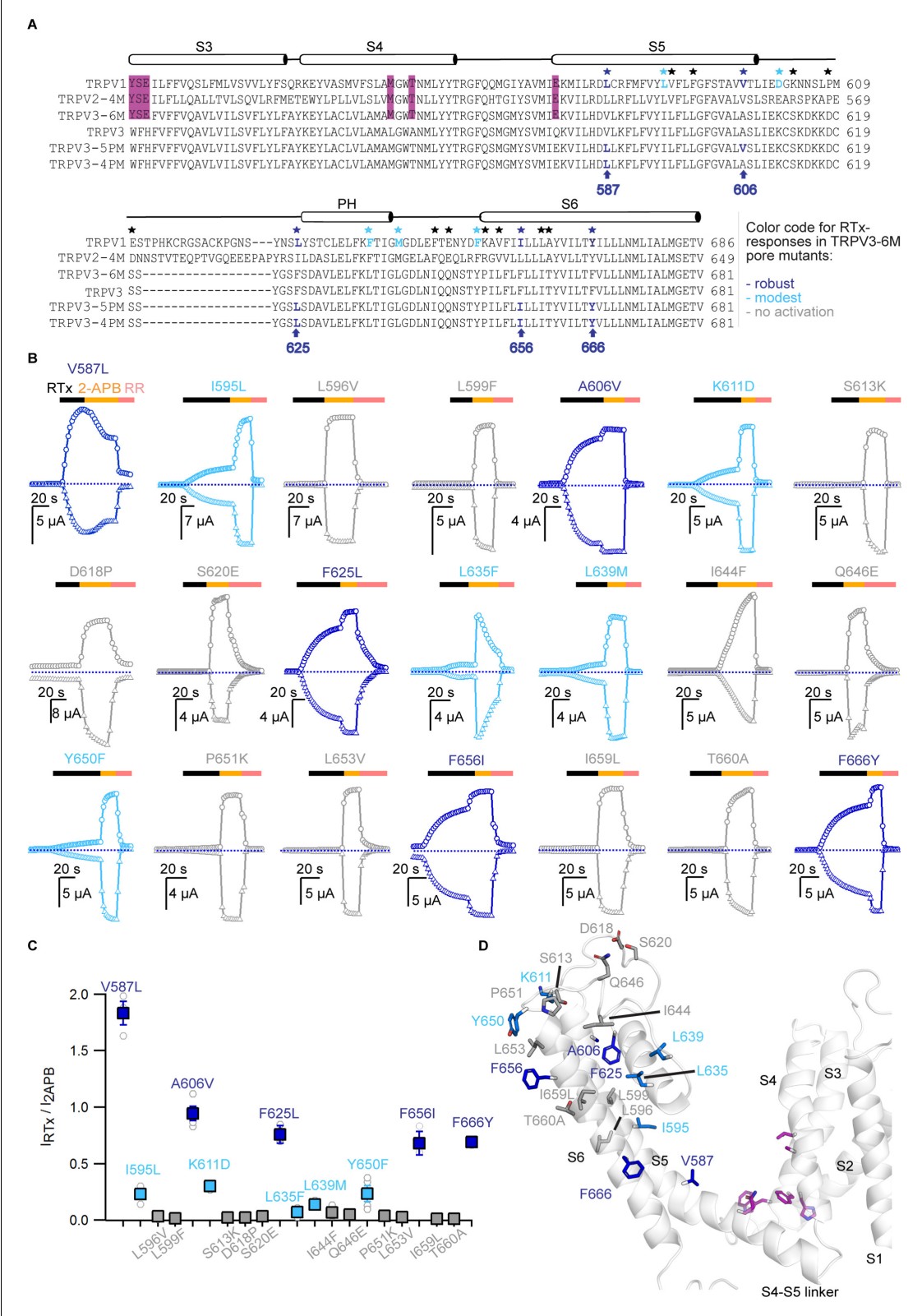

**Figure 2.** Identification of key residues for RTx activation in the pore domain of TRPV3-6M. (**A**) Sequence alignment of the S3 to S6 TM helices for rTRPV1, rTRPV2-4M, mTRPV3-6M, WT mTRPV3, TRPV3-5PM and TRPV3-4PM. Residues labeled with stars are conserved in TRPV1 and TRPV2 but different in TRPV3; black stars – mutations that did not influence RTx activation; blue - mutations that enabled moderate (light blue) or strong (dark blue) responses to RTx. The purple highlights denote the 6M mutations. (**B**) Representative time courses of activation of TRPV3-6M channels with

*Figure 2 continued on next page*

*Figure 2 continued*

individual pore mutations. Channels were stimulated by RTx (100 nM) and 2-APB (3 mM), and blocked with RR (50 μM) as indicated by the colored horizontal lines. Currents were measured at +60 (circles) and −60 mV (triangles) as in *Figure 1*. The dotted horizontal lines indicate the zero-current level. (C) Summary of the current magnitudes activated in response to RTx relative to saturating 2-APB at +60 mV from experiments as in (B). Values for individual oocytes are shown as open circles and mean ±S.E.M. as squares (n = 3–6). (D) Side view of a cartoon representation of the transmembrane domain of a mTRPV3 subunit (apo, closed structure, PDB: 6DVW) (*Singh et al., 2018*). The side-chains of residues that enable strong RTx activation when mutated are shown in dark blue, for those that enable weak RTx activation in light blue and for mutations without effect in light grey. Residues within the RTx-binding pocket that were mutated in the 6M construct are highlighted in purple. Helices are shown with 20% transparency, to visualize all side-chains.

DOI: https://doi.org/10.7554/eLife.42756.004

The following figure supplements are available for figure 2:

**Figure supplement 1.** Structural mapping of point mutations in the pore domain of TRPV3.
DOI: https://doi.org/10.7554/eLife.42756.005

**Figure supplement 2.** RTx exhibits high apparent affinity and slow dissociation.
DOI: https://doi.org/10.7554/eLife.42756.006

bound RTx and side-chains from that equivalent position in the RTx/DkTx-bound TRPV1 structure (*Figure 2—figure supplement 1C*) (*Gao et al., 2016*) or the RTx-bound TRPV2-4M structure (*Zubcevic et al., 2018b*). It seems therefore unlikely that the rest of the mutated residues that enabled RTx activation in TRPV3-6M directly interact with the vanilloid. The observation that each of the 10 identified residues enable RTx activation when mutated individually, together with their wide-spread distribution within the pore domain, suggest that state-dependent differences in the free energy contributions of individual side-chains throughout the protein can determine sensitivity to an agonist without requiring that each of these residues forms part of an interconnected allosteric network. This is similar to the hypothesis that state-dependent differences in the hydration of residues dispersed throughout the protein could potentially determine temperature-sensitivity in temperature-activated TRP channels (*Clapham and Miller, 2011*).

Estimates of the apparent affinity for RTx obtained from dose-response relations would provide an opportunity to investigate the energetic additivity in the contributions to RTx sensitivity for each of the individual pore mutations. However, RTx is very hydrophobic and likely experiences pronounced membrane partitioning and accumulation, which together with likely a high affinity for the TRPV3-6M channels results in an exceedingly slow dissociation as described previously for TRPV1 and TRPV2-4M (*Yang et al., 2016*; *Zhang et al., 2016*), and as shown for TRPV3-6M + V587L and a construct containing all 5 cumulative pore mutations in the 6M background (TRPV3-6M + 5PM) (*Figure 2—figure supplement 2*). Indeed, for these two constructs a 10-fold smaller RTx concentration (10 nM) elicited currents of similar magnitude as those activated by 100 nM RTx, but over a longer application time, which were also similar to those activated by 400 nM RTx in the same cells (*Figure 2—figure supplement 2*). Together, the very slow dissociation of RTx and the slow activation kinetics by this agonist preclude the measurement of dose-response relations or using the activation kinetics as an estimate for the association rate for RTx. Instead, we decided to investigate how these mutations impact activation of TRPV3 channels by other stimuli.

## Temperature sensitivity of TRPV3 constructs

The binding of each agonist to its site in the channel must allosterically influence the conformation of the pore domain to open the ion conduction pathway. Therefore, it is possible that the mutations in the pore that enable strong RTx-dependent activation of TRPV3 do so by shifting the gating equilibrium towards the open state, which would facilitate activation by RTx and other stimuli as well. Indeed, opening of TRPV3 upon initial stimulation seems to be largely disfavored when compared to TRPV1 and TRPV2 (*Liu et al., 2011*; *Liu and Qin, 2016*). For example, the TRPV3 channel can only initially be activated by heat at temperatures > 50°C, but will respond to much lower temperatures on subsequent heating (*Liu et al., 2011*). To test if the five pore mutations affect the overall gating equilibrium, we determined if they facilitate TRPV3 activation by heat. For these experiments we used the two-electrode voltage clamp technique together with a temperature-controlled perfusion system, as previously described (*Figure 3A*) (*Zhang et al., 2018*). First, we detected robust activation by heat in TRPV1-expressing oocytes, which we used as a positive control (*Figure 3B*). For WT

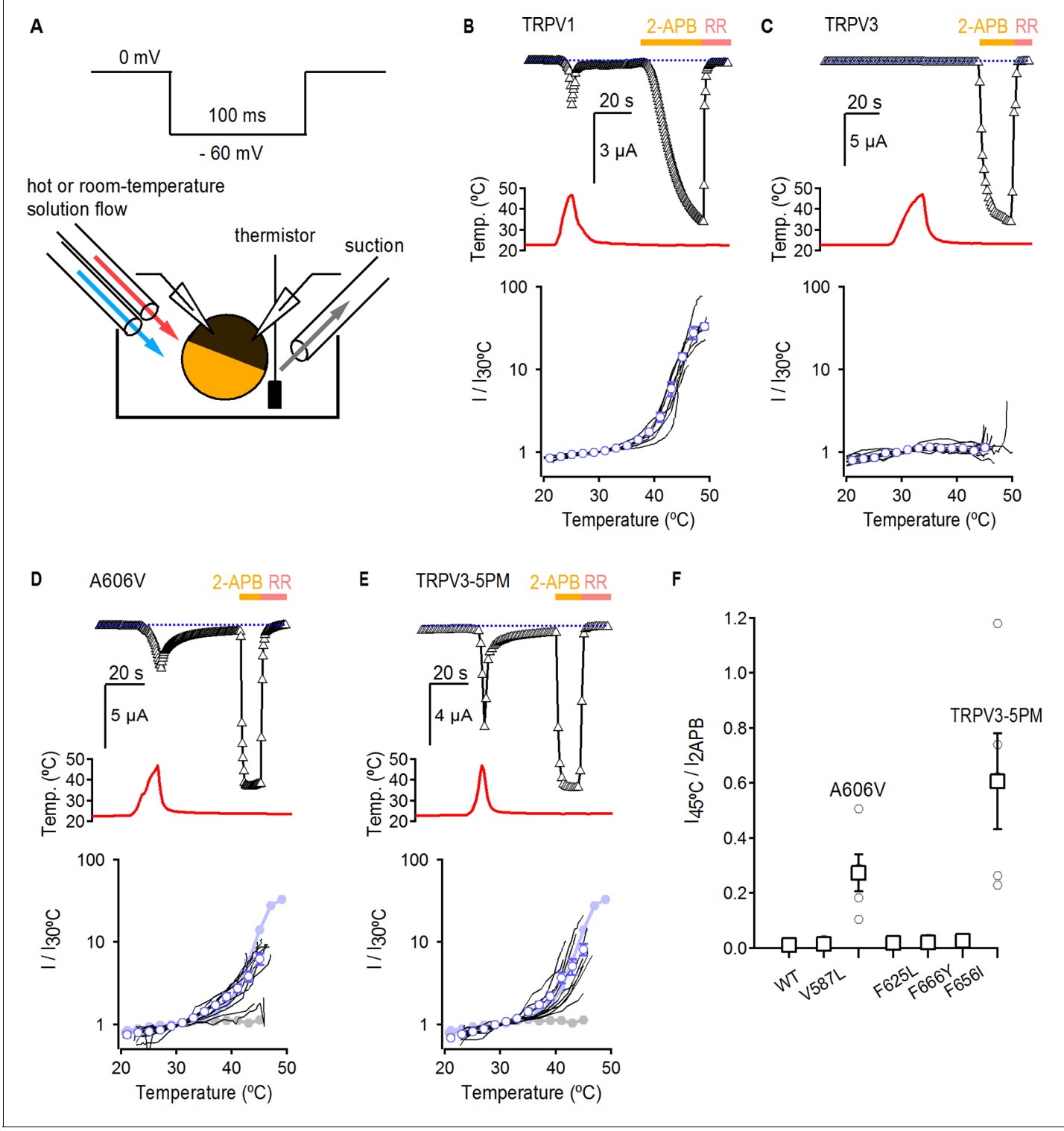

**Figure 3.** Temperature sensitivity of TRPV3 constructs. (**A**) Voltage protocol used for recording the temperature responses (top), and a cartoon of the temperature-control system used for the experiments (bottom). Temperature was controlled by using two perfusion lines immersed in baths at high- or room-temperature (*Zhang et al., 2018*), and temperature was measured with a thermistor positioned close to the oocyte. (**B**) Representative time-course obtained from a TRPV1-expressing oocyte (upper panel), showing the response to a heating stimulus, followed by the application of 3 mM 2-APB and RR (50 μM). The dotted line indicates the zero-current level. The recorded temperature is shown in the middle panel. The bottom panel shows the currents in a log-scale at −60 mV normalized to their value at 30°C plotted as a function of temperature, obtained from experiments as in the upper panel. Data from individual cells are shown as black curves, and the mean ± S.E.M as blue open circles (n = 7). (**C–E**) Representative current- (top panel)

*Figure 3 continued on next page*

*Figure 3 continued*

and temperature- (middle panel) time courses obtained from the constructs indicated. The lower panel shows the I-temperature relations in a log-scale obtained from experiments as in the upper panel (n = 5–11). In (D) and (E), the curves in solid circles correspond to the mean I vs T relations for WT TRPV1 (light blue) and TRPV3 (grey). (F) Summary of current responses to heat (45°C) relative to saturating 2-APB (3 mM) at room temperature. Values for individual oocytes are shown as open circles and mean ±S.E.M. as open squares (n = 4–5).

DOI: https://doi.org/10.7554/eLife.42756.007

The following figure supplements are available for figure 3:

**Figure supplement 1.** Temperature-activation of TRPV3 constructs.

DOI: https://doi.org/10.7554/eLife.42756.008

**Figure supplement 2.** $Q_{10}$ values.

DOI: https://doi.org/10.7554/eLife.42756.009

TRPV3, we failed to observe any response to a single heat stimulus <50°C, but observed robust responses to 2-APB (*Figure 3C,F*), consistent with previous reports (*Liu et al., 2011*; *Liu and Qin, 2017*). We observed the same result for TRPV3-6M, indicating that the 6M mutations do not noticeably influence channel activation or temperature-sensing (*Figure 3—figure supplement 1A*). We then measured the responses to heat and 2-APB of five TRPV3 constructs without the 6M mutations, each containing one of the pore mutations that gave rise to robust RTx activation in TRPV3-6M. Four out of five single point mutants (V587L, F625L, F656I and F666Y) displayed strong responses to 2-APB but no activation by temperatures below 50°C (*Figure 3F* and *Figure 31—figure supplement 1B–E*), with a heat-sensitivity of current-temperature relations that was not qualitatively different from ion diffusion as assessed from their $Q_{10}$ values (*Figure 3—figure supplement 2*). In contrast, the A606V mutant exhibited temperature-activated currents below 45°C, which were clearly larger than the responses of WT TRPV3 and TRPV3-6M, but slightly smaller than those of TRPV1 in terms of $Q_{10}$ and magnitude relative to 2-APB (*Figure 3D,F* and *Figure 3—figure supplement 2*). This result suggests that the mutation at A606 facilitates TRPV3 activation through multiple modalities, including RTx and heat. We also found that the responses to heat and the associated $Q_{10}$ values of the TRPV3/V2P chimera with and without the 6M mutations (*Figure 3—figure supplement 1F,G* and *Figure 3—figure supplement 2*), and the TRPV3-5PM construct with all five cumulative pore mutations in the WT TRPV3 background (i.e. without the 6M mutations) (*Figure 3E,F* and *Figure 3—figure supplement 2*) were qualitatively similar to that of the A606V mutant. These results suggest that the other four positions in the pore do not contribute to temperature-dependent activation. Consistently, a construct containing all pore mutations except A606V (TRPV3-4PM) exhibited no response to temperatures below 50°C (*Figure 3—figure supplement 1H* and *Figure 3—figure supplement 2*).

## Residues in the pore have distinct effects on activation by different stimuli

The results thus far indicate that A606V facilitates activation by temperature and enables RTx sensitivity, whereas the other four mutated positions in the pore domain have no effect on temperature-dependent responses. We wondered whether the mutations might allosterically favor activation by other stimuli. 2-APB is a common agonist for TRPV1, TRPV2 and TRPV3 channels, and in mouse TRPV3 it seems to bind away from any of the mutants affecting RTx sensitivity in TRPV3-6M (*Figure 2—figure supplement 1A*) (*Singh et al., 2018*), allowing us to investigate the allosteric effects of these mutants on activation by 2-APB. We therefore set out to measure the effects of pore mutations introduced into the WT background (i.e. no 6M mutations) on concentration-response relations for activation by 2-APB. We first measured the response of WT TRPV3 to 2-APB, and obtained an apparent affinity of 460 ± 12 µM (*Figure 4A,F*), consistent with a previous report (*Phelps et al., 2010*). Interestingly, the V587L mutation, which robustly promotes activation by RTx (*Figure 2B*), did not have any effect on the apparent affinity for 2-APB (440 ± 13 µM; *Figure 4B,F*). In contrast, the A606V mutation produced a detectable (~3.5 fold) increase in the apparent affinity to 132 ± 7 µM (*Figure 4C,F*), consistent with this mutation having a generalized effect on TRPV3 channel gating, as it affects the sensitivity of the channel to all tested stimuli. Interestingly, we found that a construct containing four cumulative pore mutations without A606V (TRPV3-4PM) exhibited a small ~2 fold shift in the apparent affinity for 2-APB (210 ± 2 µM, *Figure 4D,F*), which combined synergistically

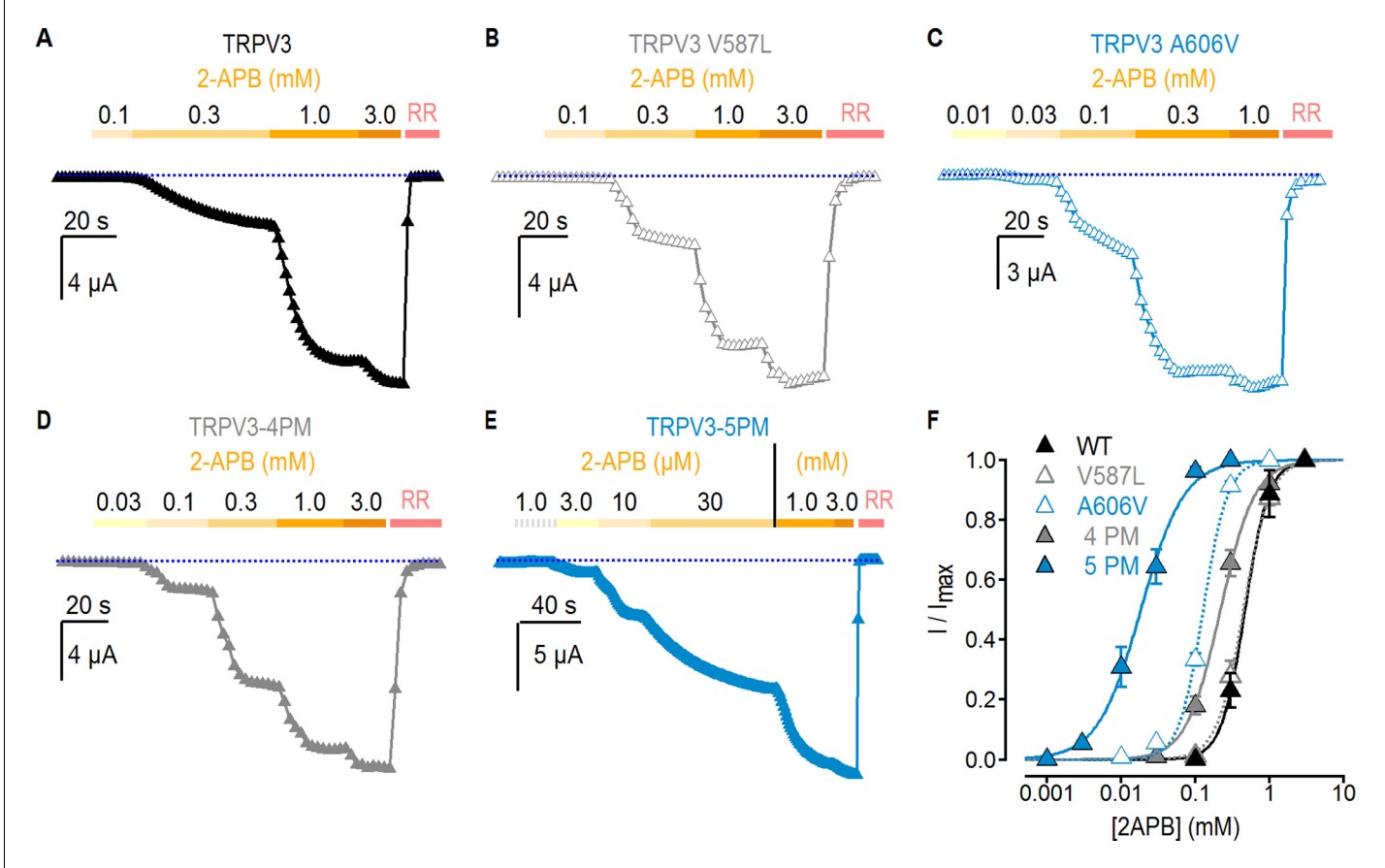

**Figure 4.** Effect of pore mutations on TRPV3 2-APB sensitivity. (**A–E**) Representative current time courses at −60 mV at increasing concentrations of 2-APB, followed by block with RR (50 μM), as indicated by the colored horizontal lines. The blue-dotted lines indicate the zero-current level. Currents were held at 0 mV and voltage was stepped to −60 mV for 100 ms every 2 s. (**F**) Mean normalized concentration-response relations for 2-APB measured from experiments as in (**A–E**). Data are shown as mean ± S.E.M. (n = 4–6). The continuous curves are fits to the Hill equation with $EC_{50}$ and s (slope) values as follows: TRPV3, $EC_{50}$ = 464 ± 12 μM, s = 2.63; TRPV3-4PM, $EC_{50}$ = 210 ± 20 μM, s = 1.84; TRPV3-A606V, $EC_{50}$ = 132 ± 7, s = 2.72; TRPV3-5PM, $EC_{50}$ = 19 ± 2 μM, s = 1.32; TRPV3-V587L, $EC_{50}$ = 443 ± 13 μM, s = 2.28.

DOI: https://doi.org/10.7554/eLife.42756.010

The following figure supplement is available for figure 4:

**Figure supplement 1.** Relative maximal activation by RTx and 2-APB for TRPV3 constructs.

DOI: https://doi.org/10.7554/eLife.42756.011

with A606V in TRPV3-5PM for a 20-fold increase in the apparent affinity (19 ± 2 μM, *Figure 4E,F*). Together, these results suggest that the five residues in the pore facilitate opening by distinct mechanisms, differentially affecting activation by heat and 2-APB, while each enabling activation by RTx. Notably, the maximal response to 2-APB was similar to the response to RTx in the 4PM and 5PM mutant channels when tested in the 6M background (TRPV3-6M/4PM and TRPV3-6M/5PM) (*Figure 4—figure supplement 1*), as well as in all five individual pore mutants when in the 6M background, with the exception of V587L, which together with L635F exhibited 2-APB-dependent desensitization and thus a higher $I_{RTx}$: $I_{2APB}$ ratio (*Figure 2B,C*).

## Heat and 2-APB render TRPV3-6M channels sensitive to RTx in the absence of mutations in the pore domain

The results thus far suggest that the A606V mutation favors channel opening in a stimulus-independent manner, and that this is sufficient to allow TRPV3-6M channels to open in response to RTx at room temperature. If this were true, the TRPV3-6M channel should be able to respond to RTx if opening is facilitated by other stimuli that activate TRPV3, such as heat or 2-APB. Consistent with

the WT TRPV3 channel being unable to bind RTx, as it lacks the 6M mutations in the binding pocket, application of the vanilloid together with heating did not elicit any response (*Figure 5A*). Consistent with our hypothesis, application of a short heating pulse together with RTx resulted in prominent activation of TRPV3-6M at temperatures only slightly above room temperature (*Figure 5B*), in contrast to the lack of response to heating up to 45°C in the absence of RTx for the same construct (*Figure 3—figure supplement 1A*). These results indicate that RTx and temperature act cooperatively to activate TRPV3-6M under conditions where neither stimulus alone would suffice. Consistent with cooperativity between heat and RTx binding, and our proposed mechanism for A606V, both the TRPV3-6M/V2P chimera and TRPV3-6M A606V responded to RTx at room temperature (*Figures 1F* and *2B*), and TRPV3 A606V was activated by a heating stimulus <50°C (*Figure 3D*). In addition, activation of the TRPV3-6M/V2P chimera to RTx observed at room temperature could be reduced to baseline levels when temperature was decreased to 10°C (*Figure 5C*), providing further evidence of coupling between heat-dependent activation and RTx sensitivity. We also examined whether we could sensitize RTx activation of TRPV3-6M by pre-stimulation with a saturating concentration of 2-APB. Similar to our results with heat, application of RTx after a short stimulation with 2-APB did not activate WT TRPV3 channels (*Figure 5D*), but resulted in robust activation of TRPV3-6M channels in the absence of additional pore mutations (*Figure 5E*). Collectively, these results demonstrate that RTx binds to the TRPV3-6M channel, but that further facilitation of channel opening is required for the vanilloid to activate TRPV3-6M.

## Discussion

The goal of the present study was to determine whether sensitivity to vanilloids could be engineered into the TRPV3 channel. Our results demonstrate that like TRPV2, the TRPV3 channel contains a 'defunct' vanilloid-binding site that can be readily made functional by mutating a few key positions in the binding pocket. More importantly, our results show that the allosteric machinery required for coupling vanilloid-binding to pore-opening has remained functional through evolution in both TRPV2 and TRPV3 channels, even though these two channels are not naturally sensitive to vanilloid molecules. This points to the presence of conserved mechanisms of gating among structurally related but functionally distinct TRP channels. In contrast to TRPV2, RTx binding to the engineered vanilloid site in TRPV3 was not sufficient to promote channel activation at room temperature, which required further manipulations: either co-stimulation with heat or sensitization with 2-APB, or the introduction of single point mutations at one of 10 identified positions throughout the pore that are conserved in TRPV1 and TRPV2 but differ in TRPV3. This indicates that agonist sensitivity in TRP channels is not only fine-tuned by the structure of the agonist-binding pocket, but also by the intrinsic energetics of activation that are influenced by multiple sites throughout the pore. Out of the five positions in the pore that enabled robust RTx activation in the TRPV3-6M construct, A606 at the extracellular end of the S5 helix had an effect on temperature-activation and moderately increased sensitivity to 2-APB. In contrast, the other four positions had no clear influence on heat-dependent activation and only produced a large (>20 fold) increase in 2-APB sensitivity when introduced cumulatively and together with A606V. These findings point to the presence of non-overlapping allosteric networks in the pore, and likely the rest of the protein, that selectively influence activation by specific stimuli.

It is intriguing that both TRPV2 and TRPV3 can respond to RTx once the vanilloid pocket has been engineered. The presence of densities near the vanilloid pocket in the cryo-EM structures of apo-TRPV1 (*Gao et al., 2016*), -TRPV2 (*Zubcevic et al., 2016*), -TRPV3 (*Zubcevic et al., 2018a*), -TRPV5 (*Hughes et al., 2018b*) and -TRPV6 (*McGoldrick et al., 2018*) have led to the hypothesis that the site can also be occupied by lipids in these channels, leading to inhibition of TRPV1 (*Gao et al., 2016*) or activation of TRPV6 (*McGoldrick et al., 2018*). Indeed, it has been shown that the endogenous monounsaturated fatty acid oleic acid inhibits TRPV1 by binding to the vanilloid pocket (*Morales-Lázaro et al., 2016*). This raises the possibility that the functional coupling between the vanilloid site and the pore is maintained in vanilloid-insensitive TRPV channels because it is important for their modulation by lipids or other endogenous modulators.

How similar are the mechanisms for RTx activation in TRPV1, TRPV2-4M and TRPV3-6M? The overall fold of the vanilloid site, located at a key interface between the S1-S4 domain and the pore (*Jordt and Julius, 2002*; *Cao et al., 2013*; *Gao et al., 2016*; *Zhang et al., 2016*; *Singh et al., 2018*;

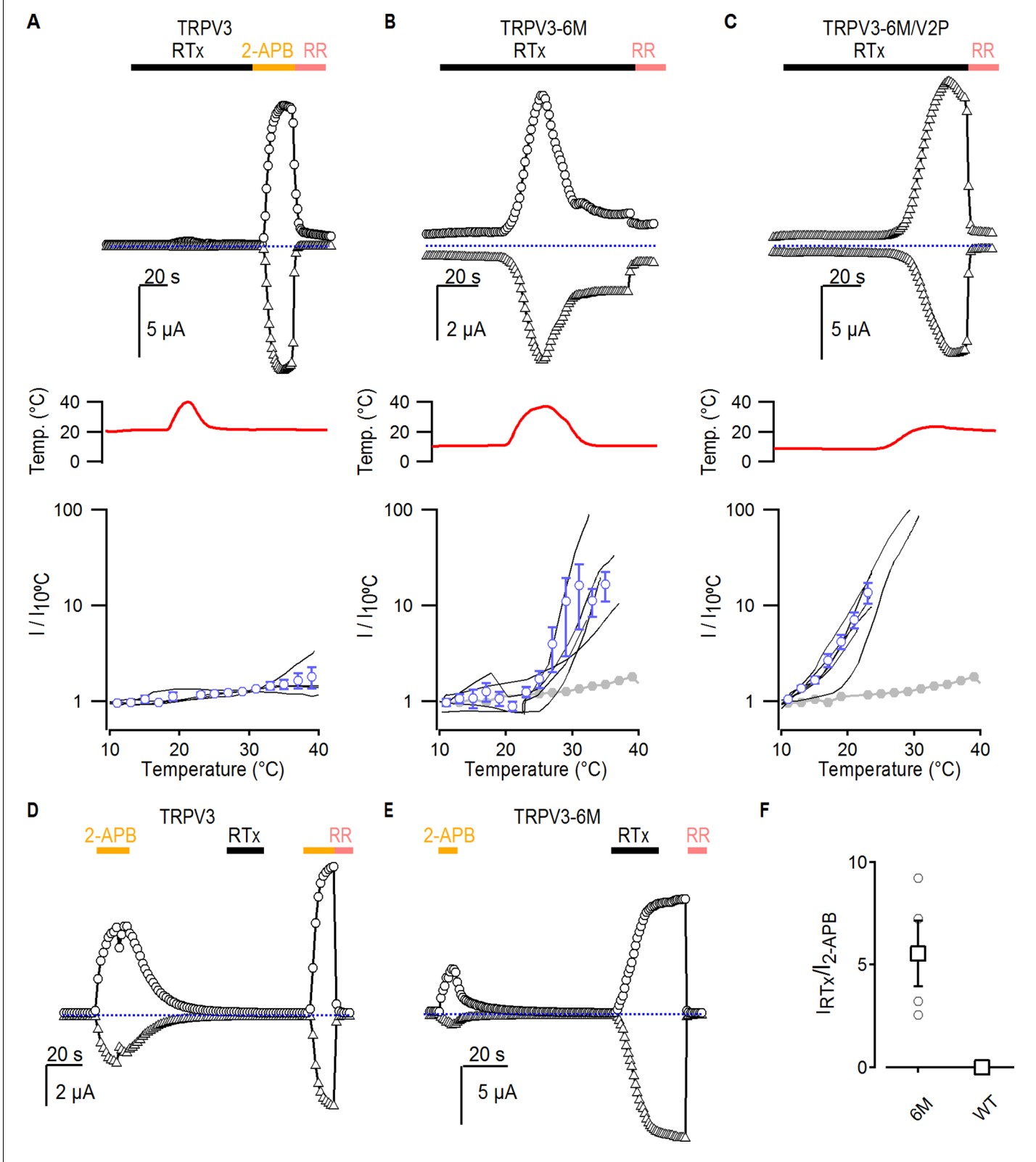

**Figure 5.** Interaction between RTx, temperature and 2-APB in TRPV3 channels. (**A–C**) Representative time courses for (**A**) WT TRPV3, (**B**) TRPV3-6M and (**C**) TRPV3-6M/V2P activation in response to 100 nM RTx and changes in temperature, which is shown in the middle panel. WT TRPV3 was also activated by 2-APB (3 mM), and all currents were blocked by RR (50 μM). Time courses were obtained at +60 (circles) and −60 mV (triangles) as described in *Figure 1B*, and temperature was controlled and monitored as in *Figure 3*. The blue-dotted horizontal lines indicate the zero-current level. The bottom

*Figure 5 continued*

panel shows normalized current vs temperature relations in a log scale at −60 mV obtained from experiments as in the upper panels. Individual cells are shown as black curves, with currents normalized to their amplitudes at 10℃, and the mean ± S.E.M. is shown in blue open circles (n = 3–4). For (B) and (C), the mean I vs T for WT TRPV3 is depicted in solid grey circles. (D, E) Representative time courses of activation of (D) WT TRPV3 and (E) TRPV3-6M in response to 2-APB (3 mM), RTx (100 nM), and RR (50 μM) at ±60 mV. (F) Summary of the current amplitude ratios for TRPV3 and TRPV3-6M activation by RTx relative to 2-APB. Values for individual oocytes are shown as open circles, the mean ± S.E.M. in open squares (n = 4–6).

DOI: https://doi.org/10.7554/eLife.42756.012

The following figure supplement is available for figure 5:

**Figure supplement 1.** Pore mutants of TRPV3 do not enable sensitivity to other TRPV1-specific agonists.

DOI: https://doi.org/10.7554/eLife.42756.013

*Zubcevic et al., 2018b*), and the overall orientation of side-chains in the binding site known to establish key interactions for activation by RTx are highly conserved between TRPV1, TRPV2 and TRPV3 channels (*Figure 1—figure supplement 1A*). Indeed, the crystal structure of the TRPV2-4M channel solved with RTx bound (*Zubcevic et al., 2018b*) suggests a similar binding pose for the vanilloid when compared to the structure of TRPV1 bound to RTx and the double knot toxin (DkTx) (*Figure 1—figure supplement 1A*). Interestingly, whereas the cryo-EM structures of closed, apo TRPV2 display four-fold (C4) symmetry relative to the central pore axis (*Huynh et al., 2016*; *Zubcevic et al., 2016*), the crystal structures of TRPV2-4M obtained in non-conducting conformations in the absence and presence of RTx (*Zubcevic et al., 2018b*) have symmetry-breaking features in the S4-S5 linker resulting in C2-symmetry for the tetrameric channel. The break in C4-symmetry was therefore proposed as an intermediate step in the mechanism of activation by RTx in TRPV2-4M (*Zubcevic et al., 2018b*). A similar symmetry-breaking step was suggested for activation of human TRPV3 by 2-APB (*Zubcevic et al., 2018a*), where a subpopulation of 2-APB-treated particles adopt non-conducting conformations with central C2-symmetry, whereas structures of a fully open, 2-APB-bound conformation of mouse TRPV3 and of non-conducting, apo states of both human and mouse TRPV3 exhibit C4 symmetry (*Singh et al., 2018*; *Zubcevic et al., 2018a*). In contrast to TRPV2 and TRPV3, no intermediate conformations with C2 symmetry have been identified for TRPV1, even if the data-set for the RTX/DkTx/TRPV1 complex is analyzed without imposing a central C4-symmetry axis (*Bae et al., 2016*; *Gao et al., 2016*). A comprehensive exploration of the conformational space of these proteins, including structures of RTx-bound TRPV1 in the absence of DkTx, and fully open RTX-bound TRPV2-4M, will be required to understand how changes in symmetry are associated with activation in each of these proteins.

Another difference between structurally-inferred activation mechanisms in TRPV1 and other TRPVs involves an α- to π-helical conversion of a segment of the S6 helix, which results in a disruption in the helix-stabilizing backbone hydrogen bonds in that segment and is proposed to function as a hinge through which the lower S6 tilts to open the internal entrance to the pore (*Palovcak et al., 2015*; *Zubcevic et al., 2016*; *Kasimova et al., 2018*; *McGoldrick et al., 2018*) (*Figure 2—figure supplement 1B,D*). The S6 helices in the non-conducting, apo structures of TRPV2 (*Huynh et al., 2016*; *Zubcevic et al., 2016*; *Zubcevic et al., 2018b*), TRPV3 (*Singh et al., 2018*; *Zubcevic et al., 2018a*) and TRPV6 (*McGoldrick et al., 2018*) are entirely α-helical, whereas the S6 helices in the open-state structures of TRPV3 and TRPV6 contain a π-helical bulge from which their inner portion tilts, opening the activation gate and exposing a different face of the helix to the central aqueous pore (*Figure 2—figure supplement 1B and D*) (*McGoldrick et al., 2018*; *Singh et al., 2018*; *Zubcevic et al., 2018a*). In the case of TRPV1, all structures in either open or non-conducting conformations contain a π-bulge in the S6 (*Figure 2—figure supplement 1C*) (*Cao et al., 2013*; *Liao et al., 2013*; *Gao et al., 2016*). More studies are therefore required to determine whether the S6 of TRPV1 also undergoes an α- to π-helix transition during activation, or whether TRPV1 contains a persistent π-helical bulge that shifts positions along the helix to enable gating (*Palovcak et al., 2015*).

It is intriguing to consider whether the other members of the TRPV family could be engineered to be activated by RTx. On the one hand, the structure of the zebrafish TRPV4 channel exhibits significant differences relative to all other TRPVs, particularly in the orientation of the S1-S4 domains relative to the pore (*Deng et al., 2018*). This results in considerable changes in the region of TRPV4 corresponding to the vanilloid pocket of TRPV1 (*Figure 1—figure supplement 1B*), making it

unlikely that RTx could bind in the state represented by the structures. On the other hand, the site corresponding to the vanilloid pocket in the structures of TRPV5 (*Hughes et al., 2018a*) and TRPV6 (*McGoldrick et al., 2018*) is very similar to that of RTx-sensitive TRPV channels (*Figure 1—figure supplement 1B*). Interestingly, non-protein densities in the vanilloid pocket observed in cryo-EM structures of TRPV5 (*Hughes et al., 2018a*) and TRPV6 (*McGoldrick et al., 2018*) have been proposed to represent either the bound inhibitor econazole or a channel-activating lipid, respectively, raising the possibility that the pocket may be functionally coupled to the pore in these channels.

It is interesting that in contrast to TRPV1 and TRPV2-4M, binding of RTx to TRPV3-6M does not readily elicit activation at room temperature. Before any previous stimulation, activation of TRPV3 seems to involve a transition with a large energy barrier, as either higher concentrations of 2-APB or temperatures > 50°C are initially required for robust activation when compared to subsequent stimulations, which elicit maximal activation at lower temperatures (<40°C) or 2-APB concentrations (*Chung et al., 2004*; *Liu et al., 2011*; *Liu and Qin, 2017*). This 'sensitization' is irreversible over the explored time-scale of several minutes, indicating that the return kinetic barrier towards the initial closed state is also high, and making it possible to shift the net apparent gating equilibrium towards the open state by either surpassing or reducing this kinetic barrier (*Liu et al., 2011*). Activation of TRPV1 and TRPV2 by heat involves significant hysteresis as well, as both the apparent threshold for activation and the steepness of the current-temperature relations are reduced after channels have been initially activated by heat (*Liu and Qin, 2016*; *Sánchez-Moreno et al., 2018*). However, in contrast to TRPV3, TRPV1 channel opening by the vanilloid capsaicin does not involve hysteresis (*Liu and Qin, 2016*), suggesting that TRPV1 activation by vanilloids does not necessarily require overcoming a large initial energy barrier. Furthermore, initial activation of TRPV1 and TRPV2 with 2-APB sensitizes both channels to subsequent 2-APB applications (*Liu and Qin, 2016*), but to a lesser degree than in TRPV3 (*Chung et al., 2004*; *Liu et al., 2011*) and more importantly, without influencing the apparent threshold or sensitivity to activation by heat (*Liu and Qin, 2016*). Together, these observations raise the possibility that a large energy barrier is an obligatory component of activation for TRPV3, whereas activation of TRPV1 and TRPV2 by some stimuli follows a path that does not involve a large kinetic barrier. The structural origin of this kinetic barrier is not known, but it is interesting to note that the open and closed TRPV3 structures exhibit far more drastic differences than those of open and closed TRPV1 (*Cao et al., 2013*; *Gao et al., 2016*) and TRPV6 (*McGoldrick et al., 2018*) structures, including significant changes in the length of the S6 and TRP helices, and a 20° rotation of the rest of the channel relative to the pore domain (*Singh et al., 2018*). Furthermore, a structure of human TRPV3 was recently obtained under conditions that favor sensitization, and it also exhibits significant conformational changes in the S6, the TRP box and the S4-S5-linker helices as compared to the apo closed state (*Zubcevic et al., 2018a*). Based in these structures it was suggested that the α- to π-transition in the S6 helix observed between the human apo and the 2-APB-sensitized structures may be part of the sensitization mechanism in TRPV3 (*Zubcevic et al., 2018a*), as no such transition has been structurally observed for TRPV1 or TRPV2 (see discussion above). It is therefore possible that RTx binding to TRPV3-6M channels is not sufficient to overcome this large energy barrier, either because the vanilloid site in TRPV3-6M may lack the allosteric connections to influence this slow transition, or because of energetically weaker coupling of the RTx site to the gating machinery. It is likely that the binding of RTx to TRPV3-6M and TRPV2-4M and/or its coupling to the pore might not function as efficiently as in TRPV1. Indeed, RTx-sensitive TRPV2 and TRPV3 constructs cannot be activated by capsaicin (*Yang et al., 2016*; *Zhang et al., 2016*) (*Figure 5—figure supplement 1A* and data not shown), a vanilloid exhibiting much lower apparent affinity when compared to RTx.

If RTx-binding to TRPV3-6M does not lead to channel activation because it is not capable of overcoming the initial energy barrier, it is expected that stimuli capable of directly activating TRPV3 channels, such as heat or 2-APB, could enable RTx activation, as we show in the present manuscript. How is RTx-sensitivity enabled in TRPV3-6M by each of the identified mutations in the pore? The A606V mutation facilitates activation by all tested stimuli, suggesting that it has a global effect on gating, possibly by favoring activation of the heat-sensing machinery, enhancing the efficacy of coupling between the stimulus-sensing domains and the pore, directly reducing the large kinetic barrier in the activation pathway and/or directly stabilizing the open state. A global effect of the mutation on gating is consistent with the lower $Q_{10}$ values for all heat-sensitive TRPV3 constructs containing the A606V mutation (*Figure 3—figure supplement 2*) relative to the $Q_{10}$ for TRPV1, as expected for

'sensitized' TRPV3 channels (*Liu et al., 2011*). This 'global' effect on gating of A606V could also explain the non-additivity in the increase in apparent affinity for 2-APB observed in TRPV3-5PM (~20 fold) relative to TRPV3-4PM (2-fold), which lacks the A606V mutation, and TRPV3-A606V (3.5-fold) (*Figure 4F*). Structurally, residue A606 at the top of the S5 helix is tightly packed against the pore helix of the same subunit (*Figure 2—figure supplement 1A,B*), so that the A606V mutation might influence the relative positioning of these two helices, resulting in a global effect on gating. It will be interesting to investigate how A606 and RTx sensitivity relate to position S404 in the membrane-proximal N-terminal domain that has been associated with temperature-dependent sensitization in TRPV3 (*Liu and Qin, 2017*).

The mechanism for the other four pore mutations that also enable robust RTx activation is more puzzling, as they all had negligible effects on activation by heat and 2-APB, suggesting that they specifically influence activation by RTx. The side-chain of V587L could potentially contribute to RTx-binding, as the residues at equivalent positions in the RTx/DkTx-bound TRPV1 (*Figure 2—figure supplement 1C*) (*Cao et al., 2013*; *Gao et al., 2016*) and RTx-bound TRPV2-4M (*Zubcevic et al., 2018b*) structures are ~4 Å from the vanilloid. However, this position has not been previously identified as an important determinant for vanilloid binding in TRPV1 (*Yang et al., 2015*; *Elokely et al., 2016*; *Zhang et al., 2016*). Importantly, V587 is located at a critical junction between the S4-S5 linker- and S5-helices that undergoes an α to π transition between the apo-TRPV2 and RTx-bound TRPV2-4M structures (*Zubcevic et al., 2018b*). This transition appears to be fundamental for coupling RTx-binding to gating in this channel, which is consistent with a key role of the S4-S5 linker in transducing the binding of capsaicin or RTx to the opening of the pore proposed for TRPV1 based on structural (*Cao et al., 2013*; *Gao et al., 2016*) and functional data (*Yang et al., 2015*; *Yang et al., 2018*). It is therefore possible that V587L specifically enhances the coupling between RTx binding and gating by influencing the conformational dynamics of the S4-S5 linker helix.

Together with V587L, mutations F625L (pore helix) and F656I (S6 helix), which promote strong responses to RTx, and I595L (S5 helix) and Y650F (S6 helix), which promote weaker responses, are all located at a protein/membrane interface that includes the vanilloid-binding pocket (*Figure 2—figure supplement 1A,B*). Multiple non-protein densities observed at that interface in the apo- or RTx/DkTx-bound structures of TRPV1 in nanodiscs have been proposed to correspond to bound lipids (*Gao et al., 2016*), which take different positions in the two structures and suggest gating-associated lipid rearrangements (*Figure 2—figure supplement 1C*). For TRPV3 it was suggested that activation by 2-APB involves the expulsion of a lipid bound near side chain Y650, based on a non-protein density close to Y650 in the apo structure of mouse TRPV3 that is absent in the 2-APB-bound open structure (*Singh et al., 2018*). Channel-lipid interactions at that interface could play a role in channel activation by RTx, raising the possibility that mutations in that interface that enable RTx activation of TRPV3-6M do so by altering protein-lipid interactions.

The lack of a global effect on gating by mutation F666Y, which enables strong responses to RTx in TRPV3-6M and is located at the π-helical portion of S6 (*Figure 2—figure supplement 1B,D*), suggests that the mutation does not largely influence the α- to π-helix transition. The equivalent residue Y671 in TRPV1 has been proposed to be responsible for coupling capsaicin-induced conformational changes in S6 to the selectivity filter and pore-helices (*Palovcak et al., 2015*; *Kasimova et al., 2018*). It is therefore possible that F666Y has a similar role in activation of TRPV3-6M by RTx, while activation by 2-APB and temperature could engage other mechanisms to influence the conformation of the filter and pore-helices. L635F and L639M also enable weak responses to RTx, and are in the pore-helix or selectivity filter, respectively, consistent with the hypothesis that conformational changes in those regions are important in the late stages of the mechanism of TRPV1 activation by capsaicin (*Kasimova et al., 2018*; *Yang et al., 2018*). However, contrary to the stimulating effect of the L635F mutation in TRPV3-6M, Phe to Leu substitutions at equivalent positions in TRPV1 (F640L) or TRPV2 (F603L) also result in potentiated responses to heat or 2-APB, respectively, and higher basal channel activity (*Myers et al., 2008*). It will be interesting to investigate the apparently opposite effects on gating that Leu or Phe side-chains have at that position in TRPV1 and TRPV2 vs TRPV3 channels. Finally, the five identified pore mutations added cumulatively to TRPV3 did not have an effect on activation by other TRPV1-specific stimuli, such as extracellular protons (*Jordt et al., 2000*) or double knot toxin (DkTx) (*Bohlen et al., 2010*; *Cao et al., 2013*; *Gao et al., 2016*) (*Figure 5—figure supplement 1B,C*), consistent with the absence of binding sites for these two modulators in TRPV3.

The fact that conservative side-chain modifications, such as V587L or I595L, can have such a dramatic effect on the response to RTx in TRPV3-6M raises the possibility that other relatively subtle structural alterations, such as those caused by the binding of another ligand, a post-translational modification, or an alteration in the properties of the membrane, could enable responses to channel modulators that would otherwise seem inactive. The picture of TRPV channel gating that emerges from this study is one where fine structural differences dictate the response of a channel to an agonist or combination of agonists. Understanding these differences might be key to unraveling the mechanisms of allosteric signal integration in TRP channels.

## Materials and methods

### Expression of constructs and molecular biology

The rat TRPV1 and TRPV2 channel (*Caterina et al., 1997*; *Caterina et al., 1999*) cDNAs were kindly provided by Dr. David Julius (UCSF), whereas mouse TRPV3 (*Peier et al., 2002*) cDNA was kindly provided by Dr. Feng Qin (SUNY Buffalo). All constructs were sub-cloned into the pGEM-HE vector for expression in oocytes. Chimeras were generated by a three-step overlapping PCR protocol and mutations were introduced using a two-step overlapping PCR mutagenesis technique. The DNA sequence of all constructs and mutants was confirmed by automated DNA sequencing and cRNA was synthesized using the T7/SP6 polymerase (mMessage mMachine kit, Ambion) after linearizing the DNA with the appropriate restriction enzymes.

### Electrophysiological recordings

All channel constructs were expressed in *Xenopus laevis* oocytes and studied using the two-electrode voltage clamp technique (TEVC) following 1–4 days of incubation after cRNA injection. The animal care and experimental procedures were performed in accordance with the Guide for the Care and Use of Laboratory Animals and were approved by the Animal Care and Use Committee of the National Institute of Neurological Disorders and Stroke (Animal protocol number 1253–17). Oocytes were removed surgically and incubated with agitation for 1 hr in a solution containing (in mM) 82.5 NaCl, 2.5 KCl, 1 MgCl$_2$, 5 HEPES, pH 7.6 (with NaOH), and collagenase (2 mg/ml; Worthington Biochemical, Lakewood, NJ). Defolliculated oocytes were injected with cRNA for each of the constructs and incubated at 17°C in a solution containing (in mM) 96 NaCl, 2 KCl, 1 MgCl$_2$, 1.8 CaCl$_2$, 5 HEPES, pH 7.6 (with NaOH), and gentamicin (50 μg/ml, GIBCO-BRL, Gaithersburg, MD). Oocyte membrane voltage was controlled using an OC-725C oocyte clamp (Warner Instruments, Hamden, CT). Data were filtered at 1–3 kHz and digitized at 20 kHz using pClamp software (Molecular Devices, Sunnyvale, CA) and a Digidata 1440A digitizer (Axon Instruments). Microelectrode resistances were 0.1–1 MΩ when filled with 3 M KCl. For recording TRP channel currents, the external recording solution contained (in mM) 100 KCl, 10 HEPES, pH 7.6 (with KOH). Experiments were performed at room temperature (~22°C), except as indicated otherwise. Agonists and RR were applied using a gravity-fed perfusion system that resulted in exchange of the 150 μL recording chamber volume within a few seconds. Current time courses were obtained by stepping membrane voltage every 2 s from a holding potential of 0 mV to either −60 or +60 mV for 100 ms, and plotting the mean steady-state current at the end of each voltage step as a function of total recording time. I-V relations were obtained similarly, but voltage was stepped every 2 s to a different value, in 10 mV increments and starting at −60 mV.

For temperature-activation experiments, calcium-activated chloride currents were minimized by using calcium-free solution containing 30 μM Caccinh-A01 and recording at negative membrane voltages. Heat stimuli were achieved by passing the external recording solution through glass capillary spirals immersed in a water bath maintained at about 70°C, and recordings were performed during constant perfusion with temperature measured using a thermistor (TA-29, Warner Instruments) located close to the cell. The thermistor was connected to the digitizer via a temperature controller (TC-324B, Warner Instruments) (*Zhang et al., 2018*). Oocytes injected with the various constructs were only studied if un-injected oocytes from the same batch contained a low density of endogenous temperature-sensitive currents at negative membrane voltages. For delivering cold stimuli, the external solution was passed through glass capillary spirals that were immersed in an ice bath. The mean I vs T relations were calculated by automatically binning each of the recordings in temperature

intervals of 2°C, and averaging a single data point within each interval per cell. All data analysis was carried out using Igor Pro 6.3 (Wavemetrics, Portland, OR). $Q_{10}$ values were obtained from $Q_{10} = 10^{10 \times m}$, where $m$ is the slope of linear fits to normalized log(current) vs temperature relations (not shown) over temperature intervals ~ 20–34°C or 38 – 45°C (*Zhang et al., 2018*).

### Reagents and chemicals
All reagents were obtained from Sigma-Aldrich (St. Louis, MO), unless indicated otherwise. Resiniferatoxin (RTx, Tocris Bioscience, Bristol, UK) and capsaicin stock solutions were made in ethanol. 2-APB stock solutions were made fresh every day in DMSO. Double knot toxin (DkTx) was synthetized in the laboratory as described previously (*Bae et al., 2012*).

## Acknowledgments

We thank Shai Silberberg, Gilman Toombes and members of the Swartz lab for helpful discussions. This work was supported by the Intramural Research Programs of the NINDS, NIH (to KJS), by an NINDS Competitive Postdoctoral Fellowship and K99 Career Development Award (to AJO).

## Additional information

### Competing interests
Kenton Jon Swartz: Reviewing editor, *eLife*. The other authors declare that no competing interests exist.

### Funding

| Funder | Grant reference number | Author |
|---|---|---|
| National Institute of Neurological Disorders and Stroke | Intramural Research Program NS002945 | Kenton Jon Swartz |
| National Institute of Neurological Disorders and Stroke | K99 Pathway to Independence Award | Andres Jara-Oseguera |

The funders had no role in study design, data collection and interpretation, or the decision to submit the work for publication.

### Author contributions
Feng Zhang, Conceptualization, Data curation, Formal analysis, Investigation, Writing—original draft, Performed experiments; Kenton Jon Swartz, Conceptualization, Resources, Supervision, Funding acquisition, Validation, Project administration, Writing—review and editing; Andres Jara-Oseguera, Conceptualization, Data curation, Software, Formal analysis, Supervision, Funding acquisition, Validation, Visualization, Methodology, Writing—original draft, Writing—review and editing

### Author ORCIDs

Kenton Jon Swartz (iD) http://orcid.org/0000-0003-3419-0765
Andres Jara-Oseguera (iD) http://orcid.org/0000-0001-5921-9320

### Ethics
Animal experimentation: The animal care and experimental procedures were performed in accordance with the Guide for the Care and Use of Laboratory Animals and were approved by the Animal Care and Use Committee of the National Institute of Neurological Disorders and Stroke (Animal protocol number 1253-17).

### Decision letter and Author response
Decision letter https://doi.org/10.7554/eLife.42756.016
Author response https://doi.org/10.7554/eLife.42756.017

## Additional files

### Supplementary files
• Transparent reporting form
DOI: https://doi.org/10.7554/eLife.42756.014

### Data availability
All data generated and analyzed for this manuscript can be visualized in the figures.

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
