## [Decision Letter]

Thank you for submitting your article "Conserved allosteric pathways for activation of TRPV3 revealed through engineering vanilloid-sensitivity" for consideration by *eLife*. Your article has been reviewed by three peer reviewers, including László Csanády as the Reviewing Editor and Reviewer #1, and the evaluation has been overseen by Richard Aldrich as the Senior Editor. The following individuals involved in the review of your submission have agreed to reveal their identity: Ramón Latorre (Reviewer #2); Seok-Yong Lee (Reviewer #3).

The reviewers have discussed the reviews with one another and the Reviewing Editor has drafted this decision to help you prepare a revised submission.

Summary:

The study by Zhang and colleagues combines electrophysiology with structure-guided mutagenesis to address the still unsolved issue of allosteric communication between stimulus sensors and gate(s) in ThermoTRP channels. As a natural extension of previous work by the same group (Zheng et al., 2016), the authors investigated to what extent the vanilloid binding site can be restored in TRPV3 channels. They show that the binding site can be engineered to bind resiniferatoxin (RTx). The vanilloid does not activate the engineered channels when applied on its own, but it does so when combined with other subthreshold stimuli which are also insufficient to open the channels on their own, such as single point mutations in the pore domain, or heating. Some of these mutations, widespread across the channel pore, decrease the temperature threshold for channel activation and also affect the sensitivity to 2-APB. These results suggest a crosstalk among these sensing modalities. The authors conclude that the specificity of activation by ligand binding is not exclusively determined by the structure of the ligand binding site, but rather by a more extended, allosteric network conserved within the TRPV family.

The reviewers find that the experiments are well designed and carefully conducted, the analysis straightforward, and that the data overall support the conclusions. However, they also find that there are a few points which require further clarifications, as detailed below.

Essential revisions:

1) Individual mutations A606V, F625L, F646I and F666Y are activated by RTx but do not reach maximal activity (they are further activated by 2-APB), but 6M-4PM and 6M-5PM constructs do. Throughout this study, the authors use a single concentration of RTx that is saturating for TRPV1. We ask the authors to provide the dose-response curves for RTX activation for the above mutants whose vanilloid sensitivity was restored. The dose-response curves could reveal to what extent the coupling between the binding site and the gate is recovered in each case, and whether the effects of the individual mutations in the 5PM construct are additive with respect to the RTx dose response.

2) Structures of human and mouse TRPV3 in multiple conformational states are available. We ask the authors to comment on how the identified mutations might be changing the energetics of channel gating, in light of those structures.

Specifically:

2.1) Are any of the mutations in the vicinity of proposed 2-APB or lipid binding sites?

2.2) Are they forming important inter- or intra-protomer contacts? It would perhaps be more informative to also show the positions of these mutations in the context of a tetrameric channel in Figure 2D. It appears that some of the residues are positioned at inter-protomer interfaces, which might indicate the mechanism by which they improve coupling?

2.3) The recent structures of mouse and human TRPV3 both show that activation of the channel results in formation of a pi-helical turn in S6, which changes the register of the helix. Does the position of any of the residues identified in this study change depending on the functional state of the channel?

2.4) The authors state that the pore mutations are all far from the vanilloid binding site, and do not interact directly with RTx. However, V587L is located at the pi-helical junction between the S4-S5 linker and S5 and is therefore close enough to be directly affected by RTx binding. This pi-helical junction was proposed as critical for allosteric coupling between the vanilloid binding site and the pore in TRPV2 (Zubcevic et al., 2018). Given that the mutation only affects the RTx response, and not heat or 2-APB it seems likely that it might have a similar role in TRPV3. Please comment.

3) TRPV3-5PM is more sensitive to temperature than the A606V single mutant (Figure 3D-E-F), whereas none of the other 4 mutants contained in 5PM seems temperature sensitive (Figure 3—figure supplement 1). It would be convenient to give the Q_10_ for all mutants to allow the reader a better quantitative grasp of how temperature sensitive the different mutants are.

4) The result obtained with the V587L mutant is intriguing. I(RTX)/I(2-APB) is ~2 for V587L (Figure 2), but ~1 for 5PM which contains mutation V587L (Figure 4—figure supplement 1). Please comment on this seemingly contradictory result.

5) The authors show that the A606V mutation on its own contributes a 3.5-fold increase in 2-APB affinity and the 4PM substitution only contributes a 2-fold increase. However, the construct combining all five mutations (5PM) increases the affinity 20-fold. Please comment on the energetic non-additivity of these effects? Have they perhaps investigated the individual mutations in the 4PM construct for their effects on 2-APB activation?

6) Although Figure 5C implicitly shows that TRPV3-6M/V2P is temperature sensitive, for the sake of completeness it would be nice to also show a simple temperature response (without RTx) for this construct (like the one shown for TRPV3/V2P in Figure 3—figure supplement 1F).

---

## [Author Response]

Essential revisions:1) Individual mutations A606V, F625L, F646I and F666Y are activated by RTx but do not reach maximal activity (they are further activated by 2-APB), but 6M-4PM and 6M-5PM constructs do. Throughout this study, the authors use a single concentration of RTx that is saturating for TRPV1. We ask the authors to provide the dose-response curves for RTX activation for the above mutants whose vanilloid sensitivity was restored. The dose-response curves could reveal to what extent the coupling between the binding site and the gate is recovered in each case, and whether the effects of the individual mutations in the 5PM construct are additive with respect to the RTx dose response.

We agree with the reviewers that having RTx dose-response relations for each of those individual pore-mutants would have provided quantitative information on the strength by which each of the mutations favors activation by the vanilloid, and also a means to test the additivity of the contributions to activation of each mutation by comparing the apparent affinity between constructs containing different combinations of mutations. Unfortunately, RTx has an exceedingly slow dissociation rate (Zhang et al., 2016, Yang et al., 2016), possibly due to very tight binding to the channels, and also because of membrane accumulation of the hydrophobic vanilloid. In addition, the kinetics of activation by RTx is slow even for saturating concentrations (see Figure 1C, F), requiring long exposure times to reach saturation and favoring membrane accumulation. Together, these properties of RTx make it impossible to have adequate experimental control of its effective concentration as applied to the channels, precluding the measurement of dose-response curves.

Instead, we have now included Figure 2—figure supplement 2, which explicitly shows the degree of reversibility of RTx-activation for two constructs, TRPV3-6M+5PM and TRPV3-6M+V587L, and that the RTx concentration used throughout the manuscript (100 nM) is saturating by comparing it to the response to a 4-fold higher concentration. In addition, in the same figure we show that a 10-fold smaller RTx concentration (10 nM) reaches the same level of activation as 100 nM during a longer exposure time, as expected for a slowly dissociating agonist with pronounced membrane partitioning. We now present these observations in the Results section (subsection “Mutations with in the pore domain required for vanilloid activation of TRPV3”, last paragraph). Finally, we think it is important to note that the responses to RTx in all five individual pore mutants were similar to those of 2-APB (Figure 2C), suggesting that RTx had a similar efficacy in all these constructs.

2) Structures of human and mouse TRPV3 in multiple conformational states are available. We ask the authors to comment on how the identified mutations might be changing the energetics of channel gating, in light of those structures.

We greatly value these suggestions, as they led to a more insightful discussion of the mechanisms by which the identified pore-mutations might influence the response of TRPV3-6M to RTX. We made a series of additions to the manuscript to address these points: First, we modified Figure 2D to include the side-chains of all mutations that had no effect on the responses to RTx, in addition to the mutations with positive effects that were initially included. Second, we have included Figure 2—figure supplement 1, which contains additional structural information to map each of the mutations in the context of the tetrameric quaternary structure, the RTx molecule, the plasma membrane, the 2-APB binding site, and the potential state-dependent conformational changes of the mutated residues as inferred from differences between TRPV3 structures in distinct functional states. The specific contents of the figure are detailed below. Third, in the Results section (subsection “Mutations with in the pore domain required for vanilloid activation of TRPV3”, first paragraph) we now provide a more detailed description of the location of each of the residues mutated in Figure 2. Fourth, the Discussion (eighth to eleventh paragraphs) now includes a section where we discuss plausible mechanisms for the effects of the identified mutants on activation of TRPV3 by RTx in the context of the structural information that is available.

Specifically:2.1) Are any of the mutations in the vicinity of proposed 2-APB or lipid binding sites?

None of the mutations are close to any of the three 2-APB binding sites proposed in the activated mouse TRPV3 structure. This can be now clearly observed in Figure 2—figure supplement 1A, which depicts the mouse TRPV3 channel tetramer showing the side-chains of all mutated residues, together with the two 2-APB molecules per subunit that are proposed to bind in the transmembrane domain. We explicitly mention this in the subsection “Residues in the

pore have distinct effects on activation by different stimuli”. In relation to the lipids, many of the mutated residues that had an effect on activation by RTx are located at a membrane/protein interface that extends to the RTx binding site, and includes many of the structural determinants proposed to be important for activation by vanilloids in TRPV1 and TRPV2-4M, such as the S4-S5 linker, the pore helix and the S5 and S6 helices (Figure 2—figure supplement 1B, C). Data from cryo-EM structures of TRPV1 (Figure 2—figure supplement 1C) and mouse TRPV3 suggest that lipids at that interface may rearrange in a state-dependent manner, such that their interactions with the channel might influence activation. In the new Figure 2—figure supplement 1B and C, we provide a structural view of this interface in TRPV3, the different conformations of the mutated residues that are within the interface in each of the available TRPV3 structures, and the proposed location of lipids in the apo and RTx/DkTx-bound structures of TRPV1 in nanodiscs. We also discuss the possible role of the lipids in explaining some of our experimental results (Discussion, ninth paragraph).

2.2) Are they forming important inter- or intra-protomer contacts? It would perhaps be more informative to also show the positions of these mutations in the context of a tetrameric channel in Figure 2D. It appears that some of the residues are positioned at inter-protomer interfaces, which might indicate the mechanism by which they improve coupling?

Figure 2—figure supplement 1A now maps all mutations in the context of the tetrameric TRPV3 structure. We did not find potential inter-protomer contacts for the mutated residues that could explain our experimental observations. We have therefore favored alternative hypotheses to explain the effects of the mutations, which are described below.

2.3) The recent structures of mouse and human TRPV3 both show that activation of the channel results in formation of a pi-helical turn in S6, which changes the register of the helix. Does the position of any of the residues identified in this study change depending on the functional state of the channel?

Interestingly, residue F666 is right at the position of the S6 helix that undergoes an α to π transition upon activation. We now clearly show this in Figure 2—figure supplement 1B, D. This change in secondary structure is associated with activation by 2-APB, such that mutations that influence the α-π equilibrium of S6 are expected to impact the sensitivity to that agonist and likely would have a global effect on gating if the α to π conversion is obligatory for TRPV3 channel opening. However, F666Y did not appreciable influence activation by temperature, and had a negligible effect on the apparent sensitivity to 2-APB when present in the TRPV3-4PM construct (Figure 4F). We therefore propose that mutation F666Y does not largely influence the α/π equilibrium of the S6 helices. We propose an alternative hypothesis based on computational and experimental observations, where F666Y could participate in the coupling of RTx-dependent conformational changes in the S6 helices to the selectivity filter and the pore helices. This is now discussed in the tenth paragraph of the Discussion. We have also included a short section on the role of the selectivity filter and pore-helices on activation of TRPV1 and TRPV2 in the context of the two mutations at those positions that enabled weak responses to RTx in TRPV36M.

The new Figure 2—figure supplement 1B and D allow readers to assess the conformation of the mutated residues as a function of the functional state of the channel, based on the available structural information for TRPV3. However, we did not identify any particular mechanism to connect the conformational differences observed in the structures with our experimental results. We have therefore provided more general hypotheses based on the overall location of the residues, and on reported structural, computational or functional observations associated with such locations.

2.4) The authors state that the pore mutations are all far from the vanilloid binding site, and do not interact directly with RTx. However, V587L is located at the pi-helical junction between the S4-S5 linker and S5 and is therefore close enough to be directly affected by RTx binding. This pi-helical junction was proposed as critical for allosteric coupling between the vanilloid binding site and the pore in TRPV2 (Zubcevic et al., 2018). Given that the mutation only affects the RTx response, and not heat or 2-APB it seems likely that it might have a similar role in TRPV3. Please comment.

The reviewers are correct that a direct interaction between the side-chain of V587L and RTx cannot be ruled out based on the available structural information, which we now clearly state in both the Results (subsection “Mutations with in the pore domain required for vanilloid activation of TRPV3”, second paragraph) and Discussion (eighth paragraph) sections. We have also included a figure of the RTx/DkTx-bound TRPV1 channel structure where the distance of the residue equivalent to V587 in TRPV1 (I577) to the RTx molecule can be visually assessed (Figure 2—figure supplement 1C, bottom panel). In addition, we now mention that V587 is located very close to the region of the S4-S5 linker helix that undergoes an α to π transition upon RTx binding in the TRPV2-4M structure, and that this change is largely responsible for the overall differences observed between the apo and the RTx-bound structures. We also refer to structural and functional data that supports a key role of the S4-S5 linker in the mechanism of activation of TRPV1 by RTx and capsaicin.

3) TRPV3-5PM is more sensitive to temperature than the A606V single mutant (Figure 3D-E-F), whereas none of the other 4 mutants contained in 5PM seems temperature sensitive (Figure 3—figure supplement 1). It would be convenient to give the Q_10_ for all mutants to allow the reader a better quantitative grasp of how temperature sensitive the different mutants are.

We have included Figure 3—figure supplement 2 that contains the Q_10_ values for all constructs tested in Figure 3 and Figure 3—figure supplement 1, measured over two different temperature ranges, one between 20-34 °C, where there is little channel activity in all constructs, and the other between 38-45 °C, over which some of the constructs exhibit pronounced activation by heat. We discuss these new data in the Results (subsection “Temperature sensitivity of TRPV3 constructs”) and Discussion sections (Discussion, seventh paragraph). We do not give a lot of emphasis to these values at a quantitative level, based on growing evidence indicating that the energetics of activation by heat in TRPV1, TRPV2 and TRPV3 channels cannot be explained by simple coupled equilibria, but that instead are associated with non-equilibrium features of channel gating (Liu et al., 2011; Jara-Oseguera et al., 2016, *eLife*; Liu and Qin, 2016; Sanchez-Moreno et al., 2018).

4) The result obtained with the V587L mutant is intriguing. I(RTX)/I(2-APB) is ~2 for V587L (Figure 2), but ~1 for 5PM which contains mutation V587L (Figure 4—figure supplement 1). Please comment on this seemingly contradictory result.

We agree with the reviewers that this result is intriguing. This is a consequence of a small 2-APB-dependent inactivation observed for TRPV3-6M + V587L and TRPV3-6M + L635F, which results in a lower steady-state value for the current relative to RTx (see the time-course for V587L and L635F in Figure 2B). Ca^2+^- and PIP_2_-independent desensitization of TRPV channels remains a poorly understood phenomenon, which seems to be tightly associated with the dynamics of the external pore and to activation by heat (Jara-Oseguera et al., 2016, *eLife*; Sanchez-Moreno et al., 2018). We therefore provide a phenomenological description of this apparently contradictory phenomenon without giving possible mechanistic explanations (subsection “Mutations with in the pore domain required for vanilloid activation of TRPV3”, first paragraph and subsection “Residues in the pore have distinct effects on activation by different stimuli”).

5) The authors show that the A606V mutation on its own contributes a 3.5-fold increase in 2-APB affinity and the 4PM substitution only contributes a 2-fold increase. However, the construct combining all five mutations (5PM) increases the affinity 20-fold. Please comment on the energetic non-additivity of these effects? Have they perhaps investigated the individual mutations in the 4PM construct for their effects on 2-APB activation?

Unfortunately we did not obtain 2-APB dose-response relations for constructs other than those included in Figure 4. The apparently global effect on gating of the A606V mutation is consistent with the non-additivity in the contributions of the other four point mutations in the context of the 5PM and 4PM constructs; if A606V favors activation by all stimuli, as our data suggests, then the effects of this mutant are expected to cause non-additive stimulating effects on other modes of activation, and on the effects of additional mutations that only influence activation by a single type of stimulus, as is the case for the other four mutants we examined. These other four mutations could potentially have additive effects on the apparent affinity for 2-APB, but the dynamic range to detect these is very narrow, as they only enhance affinity for 2-APB by a factor of 2 when all four are combined, so that it would be difficult to draw quantitative conclusions from data for the single mutants. Exploring how each of these four mutations influence sensitivity to 2-APB individually in the context of TRPV3-A606V would be interesting, but in our opinion would still not provide on their own any definite mechanistic information given that we still do not understand how A606V affects gating globally. We now discuss these observations in the seventh paragraph of the Discussion.

6) Although Figure 5C implicitly shows that TRPV3-6M/V2P is temperature sensitive, for the sake of completeness it would be nice to also show a simple temperature response (without RTx) for this construct (like the one shown for TRPV3/V2P in Figure 3—figure supplement 1F).

We have included the data for TRPV3-6M/V2P in Figure 3—figure supplement 1G and Figure 3—figure supplement 2. The data is qualitatively identical to those of TRPV3/V2P.